# Efficacy of Light-Emitting Diode-Mediated Photobiomodulation in Tendon Healing in a Murine Model

**DOI:** 10.3390/ijms26052286

**Published:** 2025-03-04

**Authors:** Jae Kyung Lim, Jae Ho Kim, Gyu Tae Park, Seung Hun Woo, Minkyoung Cho, Suk Woong Kang

**Affiliations:** 1Department of Physiology, School of Medicine, Pusan National University, Yangsan 50612, Republic of Korea; woruddla119@naver.com (J.K.L.); jhkimst@pusan.ac.kr (J.H.K.); daramzuy2@naver.com (G.T.P.); 2Department of Orthopedics, Research Institute for Convergence of Biomedical Science and Technology, Pusan National University Yangsan Hospital, School of Medicine, Pusan National University, Yangsan 56012, Republic of Korea; mudefo111@naver.com; 3Department of Parasitology and Tropical Medicine, Institute of Medical Science, College of Medicine, Gyeongsang National University, Jinju 52727, Republic of Korea

**Keywords:** photobiomodulation, light-emitting diode, tendon

## Abstract

The application of light-emitting diode (LED)-dependent photobiomodulation (PBM) in promoting post-tendon injury healing has been recently reported. Despite establishing a theoretical basis for ligament restoration through PBM, identifying effective LED wavelength combinations and ensuring safety in animal models remain unresolved challenges. In our previous study, we demonstrated that combined irradiation at 630 nm and 880 nm promotes cell proliferation and migration, which are critical processes during the early stage of tendon healing in human-derived tendon fibroblasts. Based on this, we hypothesized that 630/880 nm LED-based PBM might promote rapid healing during the initial phase of tendon healing, and we aimed to analyze the results after PBM treatment in a murine model. Migration kinetics were analyzed at two specific wavelengths: 630 and 880 nm. The Achilles tendon in the hind limbs of Balb/c mice was severed by Achilles tendon transection. Subsequently, the mice were randomized into LED non-irradiation and LED irradiation groups. Mice with intact tendons were employed as healthy controls. The total number of mice was 13 for the healthy and injured groups and 14 for the LED-irradiated injured group, and the data presented in this manuscript were obtained from one representative experiment (*n* = 4–5 per group). The wounds were LED-irradiated for 20 min daily for two days. Histological properties, tendon healing mediators, and inflammatory mediators were screened on day 14. The roundness of the nuclei and fiber structure, indicating the degree of infiltrated inflammatory cells and severity of fiber fragmentation, respectively, were lower in the LED irradiation group than in the LED non-irradiation group. Immunohistochemical analysis depicted an increase in tenocytes (SCX^+^ cells) and recovery of wounds with reduced fibrosis (lower collagen 3 and TGF-β1) in the LED irradiation group during healing; conversely, the LED non-irradiation group exhibited tissue fibrosis. Overall, the ratio of M2 macrophages to total macrophages in the LED irradiation group was higher than that in the injured group. LED-based PBM in the Achilles tendon rupture murine model facilitated a rapid restoration of histological and immunochemical outcomes. These findings suggest that LED-based PBM presents remarkable potential as an adjunct therapeutic approach for tendon healing and warrants further research to standardize various parameters to advance and establish it as a reliable treatment regimen.

## 1. Introduction

In the 1960s, Mester documented the potential applicability of photobiomodulation (PBM) in accelerating wound restoration, following which researchers at the National Aeronautics and Space Administration implemented it as a therapeutic regimen to enhance the healing process in space [1,2]. PBM facilitates healing by stimulating the mitochondrial and cell membrane photoreceptor-based synthesis of Adenosine Triphosphate (ATP), which increases cell viability [3,4]. PBM is performed using lasers or light-emitting diodes (LEDs) that radiate light in the red and near-infrared wavelengths [5,6]. Numerous studies and trials have been conducted over the past two decades on the clinical application of PBM in the medical and dental fields. PBM is employed in various disciplines of clinical dentistry for post-orthodontic treatment pain alleviation, osseointegration, collagen deposition, and implant stability enhancement [7,8,9]. PBM has demonstrated additional efficacy in various dermatological interventions, including skin rejuvenation, hair growth, and fat reduction [10,11]. Moreover, empirical evidence demonstrating the potential of PBM in promoting fibroblast proliferation, growth factor synthesis, collagen production, and angiogenesis has prompted numerous animal experiments and clinical studies in the field of orthopedic surgery [3,5,12,13]. Nevertheless, specifications on parameters such as wavelength, intensity, and irradiation time associated with PBM therapy have not yet been established, posing limitations on its clinical applicability. Drawing upon prior research, Vinck et al. reported that both LED and laser stimulate fibroblasts in the Achilles tendon of a rat model [14]. Our previous investigation also assessed the efficacy of combined 630/880 nm wavelength PBM on human tendon-derived fibroblasts and revealed an over 2-fold increase in cell proliferation and a 3-fold increase in cell migration in the PBM-treated group compared to the control group [15]. Based on this, we hypothesized that 630/880 nm PBM could promote rapid recovery during the early tendon healing phase. In this study, we aimed to evaluate the effects of 630/880 nm LED irradiation on the initial healing phase of injured Achilles tendons in a murine model.

Diseases affecting tendons and ligaments constitute a substantial proportion of orthopedic ailments [16]. The prevalence of these diseases is rising annually owing to the ongoing growth of elderly and athletic populations; overall, this trend has been associated with high social costs [17,18,19]. Many treatments are being introduced for these tendon and ligament diseases, including PBM. Although the efficacy of PBM has been established theoretically, its applicability remains to be validated in a practical or clinical context. Therefore, the aim of this study was to establish optimal PBM treatment conditions for tendon healing, a critical step toward advancing therapeutic approaches for tendon injuries. This study has significant potential to contribute to the development of effective clinical strategies, addressing a major challenge in the field of orthopedics.

## 2. Results

### 2.1. LED Irradiation Accelerated the Healing of the Achilles Tendon Injury

To investigate the effects of light-emitting diodes (LED) irradiation on Achilles tendon injury repair in vivo, we used a mouse model. Surgical transection lesions of the Achilles tendon were produced in the hind limbs of mice, and the surgical site was irradiated locally with LED, as described in the preceding Materials and Methods section. As a control, the injured Achilles tendon was not irradiated with LED. Hematoxylin and eosin (H&E) staining was performed to examine whether LED irradiation had a therapeutic effect on Achilles tendon injuries. The lesions could be clearly identified by using H&E staining of the injured Achilles tendon. LED irradiation improved tendon healing without any detectable local adverse effects (Figure 1A). Next, we analyzed the fiber structure, fiber arrangement, cell density, and roundness of the nuclei in tendon tissues. All four indicators were markedly elevated in the injury group compared to those in the normal group, and fiber structure and arrangement were notably decreased in the LED treatment group compared to the injury group (Figure 1B,C). However, there were no discernible differences in the cell density or roundness of the nuclei of injured tendons between the control and LED-irradiated mice (Figure 1D,E). These results suggest that LED treatment promotes fiber regeneration and arrangement, although the infiltration of inflammatory immune cells in the injured tendons remained unaffected

### 2.2. LED Irradiation Promoted Tenocyte Proliferation

Tenocytes are the principal cellular constituents of the tendon and play various roles during tendon injury. Tenocytes express tenomodulin (Tnmd) and scleraxis (SCX), which are well-known tenocyte markers. To explore the effects of LED irradiation on tenocyte proliferation during tendon repair, we determined the effect of LED irradiation on the number of tenocytes in the injured tendons. Immunohistochemistry of tendon tissues revealed overexpression of both genes in the injury group compared to the normal group and decreased expression in the LED-treated group (Figure 2A). Western blot analysis confirmed that the expression levels of Tnmd and SCX were higher in the injury group than in the normal group, whereas they were strikingly lower in the LED group than in the injury group (Figure 2B,C). The histological results showed that LED treatment promoted fiber regeneration and arrangement, confirming the restorative effect of LED. Moreover, on day 14, tenocyte markers decreased and were quickly normalized.

### 2.3. LED Irradiation Increased Collagen 1/3 Expression

To explore the effects of LED irradiation on collagen synthesis, we examined the expression levels of collagens 1 and 3 by using immunocytochemical and Western blot analyses. Immunocytochemical analysis exhibited that the expression levels of collagens 1 and 3 were notably higher in the injury group than in the normal group and markedly lower in the LED treatment group than in the injury group (Figure 3A). Western blot analysis demonstrated that the protein levels of both collagen 1 and 3 increased in injured tendons compared to normal tendons, and the expression levels of collagen 3 were discernibly decreased upon LED treatment in injured tendons (Figure 3B–D). The ratio of collagen 1/3 was slightly lower in the injury group compared to the normal group but was markedly restored in the LED-treated group to a level comparable to that observed in the normal group (Figure 3E). In conjunction with the increased collagen arrangement and structure (Figure 1), these results suggest that LED irradiation stimulates the organization and arrangement of collagen to promote repair while ensuring reduced inflammatory immune cell infiltration in Achilles tendon injury.

### 2.4. LED Irradiation Reduced the Degree of Fibrosis

Since LED irradiation rapidly stabilized the repairing site by regulating collagen synthesis and cross-link formation in the injured Achilles tendon, we examined the expression level of transforming growth factor beta-1 (TGF-β1) in the injured tendon using immunocytochemistry and Western blot analysis. Immunocytochemical analysis evidenced that TGF-β1 expression was increased in the injured tendon compared to the normal tendon, and it was markedly reduced by LED treatment (Figure 4A). The expression levels of TGF-β1 and vimentin increased in injured tendon tissue, and the increased expression of TGF-β1 and vimentin was considerably decreased in the LED-treated tendon, suggesting that LED irradiation may reduce fibrosis (Figure 4B–D).

### 2.5. LED Irradiation Prevented Inflammatory Macrophage Infiltration in the Injured Site

To explore whether LED irradiation affected the activation of macrophages in injured Achilles tendons, tendon tissues were immunostained for CD68, a pan-macrophage marker, and CD163, an M2 macrophage marker (Figure 5A). The experimental findings demonstrated that the number of macrophages was higher in the injury group than in the normal group; however, the number of macrophages in the injured tendon decreased in response to LED treatment (Figure 5B). The number of CD68^+^CD163^+^ M2 macrophages was higher in the injury group than in the normal group but was not notably affected by LED irradiation (Figure 5C). Nonetheless, the ratio of M2 macrophages to total macrophages was higher in the LED irradiation group than in the injured group (Figure 5D). This result suggests that LED irradiation modulates inflammatory response to promote the recovery of the injured Achilles tendon.

## 3. Discussion

The results of this study confirmed the initial effect of light-emitting diode (LED) treatment on tendon healing. The histology scores and immunochemistry results exhibited remarkable differences between the LED treatment and control groups.

Photobiomodulation (PBM) using low-level laser therapy (LLLT) or LED therapy was first introduced by Mester in the early 1960s. PBM accelerates healing by stimulating mitochondrial and cell membrane photoreceptor synthesis of ATP, thereby increasing cell activity. Such modulations in these cells can promote fibroblast proliferation, growth factor synthesis, collagen production, and angiogenesis. Various laboratory studies and animal experiments have been conducted in the field of orthopedic surgery [13,20,21,22,23,24,25,26,27,28,29,30,31,32,33,34]. Silva et al. reported a positive effect of LLLT/LED irradiation on tendon damage [25]. Rosso et al. documented that PBM has beneficial effects on the recovery of nerve lesions. In vitro experimental findings suggest that PBM may facilitate tissue homeostasis, thus stimulating articular tissue components and promoting chondroprotective effects [13].

Accordingly, various cell and animal experiments have revealed positive results for PBM in the field of orthopedic surgery, especially in the treatment of ligaments [15,25,26,27,28,29,30,31,32,33,34]. However, the optimal PBM conditions for injured tendon recovery have not been established yet. While some studies have investigated tendon healing using far-infrared radiation, research on the tendon healing effect of near-infrared radiation or co-exposure of both wavelengths in animal models has rarely been attempted. Therefore, future studies are needed to assess clinical outcomes related to this approach. In this regard, we previously demonstrated the striking effect of LED irradiation at 630 nm and 630 nm + 880 nm for 20 min on the proliferation and migration of human tendon fibroblasts [15]. Based on these promising in vitro results, we attempted to clarify whether the combined LED exposure of 630/880 nm exerts a beneficial effect on tendon healing in a murine model of Achilles tendon injury.

Ruptured Achilles tendons have been reported to exhibit marked collagen degeneration, disordered arrangement of collagen fibers, augmented cellularity, and an increase in the number of tenocytes with round nuclei [35,36]. The failure of intact tendon healing due to fibrotic scar formation after medical and surgical treatment, which can lead to chronic symptoms and reinjury, is a common issue [1,2,3,4]. Histologic analysis showed that PBM therapy improved tendon healing through inhibition of cell density and nuclear circularity. This outcome accelerated fiber regeneration and alignment, and collagen synthesis. The tissue structure in LED-irradiated injured Achilles tendons exhibited greater parallelism, in addition to having a denser deposition of new collagen fibers compared to non-irradiated injured tendons. These results indicate that LED therapy can shorten tendon recovery time, contribute to the establishment of novel treatment protocols for PBM-based tendon therapy, and help define new therapeutic conditions applicable in clinical settings.

Collagen 1 is the main ECM component of tendons, whereas collagen 3, which is generally associated with scar tissue and injury, accumulates at injured tendons. The increased content of type 3 collagen can cause thinner collagen fibers and decrease the tensile strength [37,38]. The collagen 1/collagen 3 ratio was markedly restored in the LED-treated group to a level similar to the normal group. Histological results further support that LED irradiation stimulates the organization and arrangement of collagen to promote healing.

Tendon injury is associated with tissue regeneration and fibrosis. TGF-β1 is activated upon tendon injury and is key in tendon healing and fibrosis [39,40,41,42]. The expression of TGF-β1 and vimentin, markers of the presence of myofibroblasts, has been implicated in fibrosis [23]. TGF-β1 and vimentin levels decreased to the normal range in the LED group at the two-week mark. Although our results do not directly reflect changes in collagen levels during the early stages of healing, considering the histological and collagen level results, it is thought that the levels normalized as recovery rapidly progressed.

Macrophages are key regulators of the healing of injured tendons. For example, an increase in the concentration of macrophages has been reported to play a key role in regulating the healing process of injured tendons [36,37,38,43,44]. The specific function of macrophages depends on their phenotype. While the M1 phenotype macrophage exhibits a phagocytic and proinflammatory function [6,7,8,9], the M2 phenotype macrophage is associated with tissue repair and deposition in inflamed tissue [6,7,10]. During tendon healing, increases in the concentration of the M2 macrophage phenotype occur later in the healing process [36,37,38].

In this study, the ratio of M2 macrophages to total macrophages increased in the LED irradiation group compared with that in the injured group. This increase, driven by LED irradiation, may inhibit abnormal or excessive inflammatory responses. Therefore, LED-facilitated recovery of injured Achilles tendons is controlled through an increased differentiation toward the M2 macrophage phenotype.

A systematic review examined the effects of LLLT and PBMT using LEDs on calcaneal tendon injuries in rats [18]. This study evaluated various research works to determine the effectiveness of PBMT in accelerating tendon healing, reducing inflammation, and improving tissue repair. Most studies used light in the infrared range (660–945 nm) at power levels ranging from 22 to 100 mW [13,25,26]. Energy densities varied, averaging 45.6 J/cm² for infrared and 76 J/cm² for red light. Treatment durations ranged from 3 to 170 s, depending on the study. PBMT, particularly at specific infrared and red wavelengths, can accelerate tissue repair and modulate inflammatory responses in calcaneal tendon injuries in rats. However, the effects observed in tendon injury treatment in animal models, especially rats, are difficult to directly apply to humans. Because the physiological characteristics and healing processes of animal and human tissues are different, additional clinical studies are essential to generalize the treatment results. In addition, many studies have been conducted over a short experimental period (hours to days), making it difficult to verify long-term effects [27,28,29]. In particular, since the tendon injury healing process involves multiple stages, such as initial inflammation, proliferation, and remodeling, long-term follow-up studies are necessary.

This study had several limitations. First, it was an in vivo study, and owing to its limited sample size (13–14 mice per group) and brief duration (within 2 weeks), it was difficult to fully capture the long-term healing process. Second, the mechanism underlying each stage of the healing process could not be ascertained. Therefore, it is imperative to conduct further investigations into the mechanisms and impacts of each process through long-term experiments. Another major limitation of our study lies in the fact that we were not able to conduct mechanical testing on animals. In general, mechanical testing should be performed in conjunction with histology to accurately assess the healing trajectory (or lack thereof) of a tendon. To address this limitation in future studies, we intend to conduct mechanical testing after PBM irradiation to better characterize the dynamics of tendon healing. Finally, the genetic variability of the mice and other experimental conditions, such as anesthesia, may potentially influence healing outcomes.

## 4. Materials and Methods

### 4.1. Animal Experiments

The animals were housed in an air-conditioned animal room with constant relative humidity and provided with a standard laboratory diet and water as outlined in the *Guide for the Care and Use of Laboratory Animals*. Animal experiments were performed in adherence to the protocols approved by the Pusan National University Institutional Animal Care and Use Committee (PNU-2023-0269). Six-week-old male BALB/c mice weighing 22–24 g were acquired from Koatech (Gyeonggi-do, Republic of Korea). To induce Achilles tendon injury, the mice were administered intraperitoneal anesthesia with 1.25% avertin (2,2,2-tribromoethanol, 250 mg/kg) (Sigma-Aldrich, Incheon, Republic of Korea). The Achilles tendon was exposed through an approximately 10 mm longitudinal skin incision on the medial side of the right hind limb. A transverse transection was performed with a diameter of 2 mm at the midpoint of the tendon. A monofilament suture with a knotted end was passed through the proximal stump of the severed tendon and subsequently through the distal stump to approximate the two ends. After tendon alignment, the skin incision was closed. The detailed surgical procedure is illustrated in Appendix A. This surgical procedure was slightly modified based on previously established Achilles tendon injury models [45,46]. In addition, to eliminate the effects of anesthesia as a confounding factor, the healthy control group was subjected to the same anesthesia conditions as the tendon injury group. The mice were randomized into two groups: light-emitting diode (LED) non-irradiated and LED irradiated. Mice with intact tendons were employed as healthy controls. The mice were euthanized 2 weeks following surgery, and tendon tissues were harvested and stored at −80 °C or fixed in acetone. Data presented in this manuscript were obtained from one representative experiment. In total, 13 mice were assigned to the healthy control group, 13 to the injury group, and 14 to the LED-irradiated injury group. To ensure reproducibility and reliability, the animal experiments were conducted independently three times, with 4 to 5 mice per group in each experiment.

### 4.2. LED Irradiation

The LED-irradiated group was anesthetized and subjected to wound irradiation. The LED was applied to the area of Achilles tendon injury for 20 min per session at 24 h intervals over a period of 2 weeks, commencing 24 h post-surgery. LED irradiation was applied at wavelengths of 630 nm (10 mW/cm^2^, 100 Hz) and 880 nm (40 mW/cm^2^, 100 Hz), delivering energy densities of 12 J/cm^2^ for 630 nm and 48 J/cm^2^ for 880 nm per irradiation session, respectively. The LED probe was positioned 0.5 cm above the tendon injury sites, ensuring consistent energy delivery across an irradiated area of about 3.5 cm^2^.

The LED devices were designed and manufactured for cell and animal experiments, capable of emitting light at 630 and 880 nm (Daram Inc., yangsan-si, Republic of Korea). They were designed to operate within the intensity range of 10 to 100 mW/cm^2^ and the frequency range of 10 to 100 Hz. During the experiments, the wavelength, intensity, and frequency were adjusted.

To control for anesthesia as a potential confounder, the two non-irradiated groups were also anesthetized under the same conditions as the LED-irradiated group but did not receive LED treatment.

### 4.3. Histological Analysis

The mice were euthanized, their Achilles tendons excised, and the tissue specimens were fixed overnight in acetone at −20 °C and embedded in an optimum cutting temperature (OCT) compound (Sakura Finetek USA, Inc., Torrance, CA, USA). Tissues were sectioned into 10 μm segments and stained with hematoxylin and eosin (H&E) for histological score analysis. Stained tissue sections were scanned using an Axio Scan Z1 (Carl Zeiss, Heidenheim, Germany) at ×200 magnification. The histological outcomes were scored from 0 (best) to 3 (worst) and evaluated on four parameters: cell density, roundness of nuclei, fiber structure, and fiber arrangement. Two blinded pathologists independently graded the histological results from 0 to 3, subjectively categorizing the tissue samples according to the four aforementioned parameters. Three sections were randomly selected from each sample, and the average scores for each group were compared.

### 4.4. Immunohistochemistry

Tendon sections fixed in acetone and embedded in OCT compound were incubated with anti-tenomodulin (Tnmd) (bs-7525R; Bioss, Woburn, MA, USA) and anti-scleraxis antibodies (MBS9612052; MyBioSource, San Diego, CA, USA) to evaluate tenocyte proliferation. Additionally, the collagen in the specimens was stained with anti-collagen1 (bs-10423R; Bioss, MA, USA) and anti-collagen3 antibodies (ab7778; Abcam, Cambridge, UK). The tendon sections were stained with anti-TGF-β1 antibody (MAB240-100; R&D Systems, Minneapolis, MN, USA) to examine for fibrosis and with anti-CD68 (14-0681-82; Invitrogen, Waltham, MA, USA) and anti-CD163 (ab182422; Abcam, Cambridge, UK) antibodies to identify M2 macrophages. Subsequently, the specimens were incubated with secondary antibodies (Alexa Fluor 488, 568, or 647; All from Invitrogen, Waltham, MA, USA) for 2 h at room temperature, washed, and mounted with a prolonged gold antifade mounting solution. Stained sections were visualized under a confocal microscope (LSM900, Carl Zeiss, Heidenheim, Germany). ImageJ version 1.53 software (LOCI, University of Wisconsin, Madison, MD, USA) was used to quantify the number of Tnmd^+^, SCX^+^, CD68^+^, and M2 macrophages (CD68^+^CD163^+^) in high-power fields.

### 4.5. Western Blot Analysis

Tendon tissues were homogenized and lysed using lysis buffer (pH 7.4; 20 mM Tris-HCl, 1 mM EGTA, 1 mM EDTA, 10 mM NaCl, 0.1 mM phenylmethyl sulfonyl fluoride, 1 mM Na_3_VO_4_, 30 mM sodium pyrophosphate, 25 mM β-glycerol phosphate, and 1% Triton X-100) containing protease inhibitors. The protein fractions in the lysates were resolved via sodium dodecyl sulfate-polyacrylamide gel electrophoresis and then transferred to nitrocellulose membranes (Amersham, Bensheim, Germany). Proteins were stained with 0.1% Ponceau S solution (Sigma-Aldrich Co. Ltd., St. Louis, MO, USA) and blocked with 5% nonfat milk (BD, New York, NY, USA). The bound antibodies were visualized using the corresponding horseradish peroxidase-conjugated secondary antibodies (ab6802 and 115-035-146; Abcam, Cambridge, UK and Jackson, PA, USA). The enhanced chemiluminescence Western blotting system (GE-RPN2106: Cytiva, Incheon, Republic of Korea) was used for signal detection, and images were captured using an ImageQuant 800 Western blot imaging system (GE Healthcare, Chicago, IL, USA).

### 4.6. Statistical Analysis

Results from multiple observations are presented as mean ± SD. A Student’s two-tailed unpaired *t*-test was used to determine the statistical significance of differences between the two groups. Differences between groups were evaluated using one- or two-way analysis of variance (ANOVA), followed by Scheffé’s test.

## 5. Conclusions

Histological and immunochemical outcomes demonstrated the effectiveness of LED-based PBM in stimulating rapid recovery in a murine model of Achilles tendon rupture. These results suggest that LED-mediated PBM has considerable potential as an adjunct treatment for tendon healing and warrants further research to standardize various parameters for its development and establishment as a reliable treatment regimen.

## Figures and Tables

**Figure 1 ijms-26-02286-f001:**
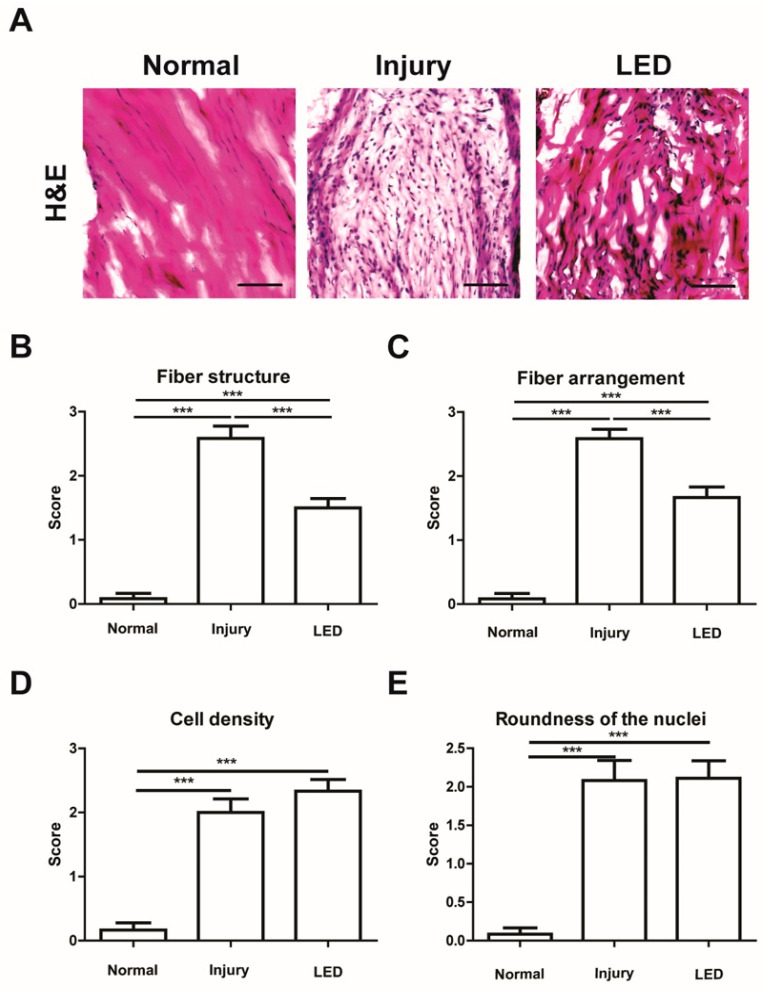
Effects of light-emitting diodes (LED) irradiation on fibrosis in injured Achilles tendon. (**A**) Hematoxylin and eosin (H&E) staining of the Achilles tendon tissue sections at 2 weeks post-surgery. Scale bar = 100 μm. (**B**–**E**) Histological analysis of the representative H&E-stained tissue section of the Achilles tendon. Fiber structure, fiber arrangement, cell density, and roundness of nuclei were quantified. *** *p* < 0.005. Data indicate mean ± SD (*n* = 4 per group).

**Figure 2 ijms-26-02286-f002:**
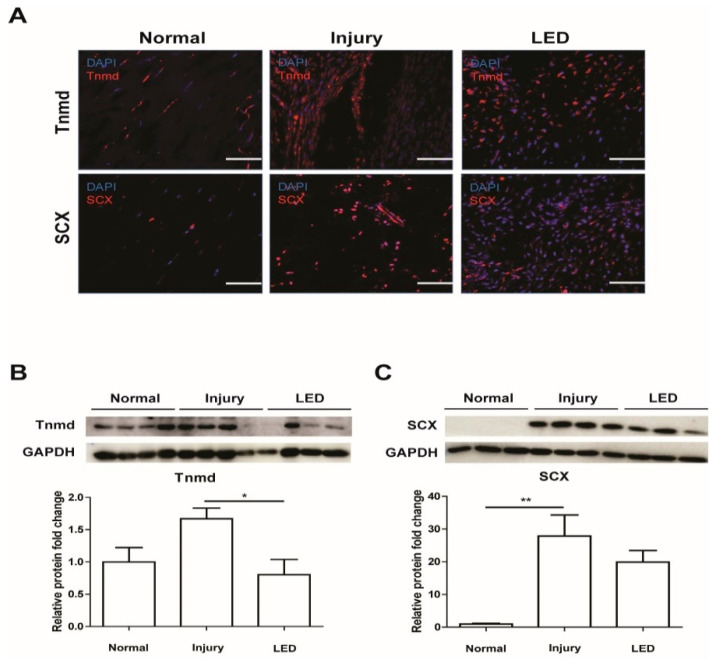
Effects of LED irradiation on tenocyte proliferation of injured Achilles tendon. (**A**) Immunostaining of Tnmd and SCX in injured Achilles tendon after 2 weeks post-surgery. Nuclei were stained with 4’,6-diamidino-2-phenylindole (DAPI). Scale bar = 50 μm. (**B**) Representative image (upper panel) of Western blot analysis of tenomodulin (Tnmd) and relative protein levels (lower panel) of Tnmd vs. glyceraldehyde-3-phosphate dehydrogenase (GAPDH) expression. (**C**) Representative image (upper panel) of Western blot analysis of scleraxis (SCX) and relative protein levels (lower panel) of SCX vs. GAPDH expression. * *p* < 0.05, ** *p* < 0.01. Data indicate mean ± SD (*n* = 4 per group).

**Figure 3 ijms-26-02286-f003:**
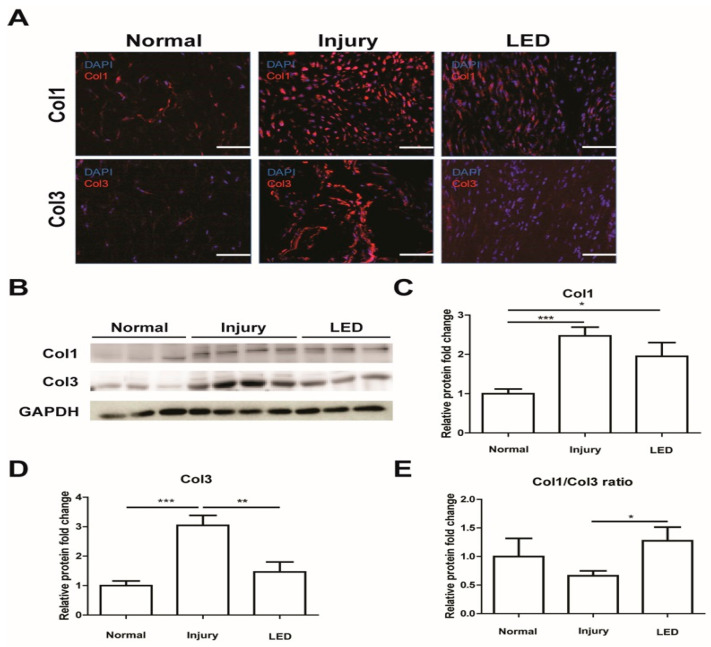
Effects of LED irradiation on collagen synthesis in injured Achilles tendon. (**A**) Immunostaining for collagen 1 and 3 in Achilles tendon tissue at 2 weeks post-surgery. Nuclei were stained with DAPI. Scale bar = 50 μm. (**B**) Representative images of Western blot analysis of collagen 1 and 3. Relative protein levels of collagen 1 (**C**) and collagen 3 (**D**) vs. GAPDH expression. (**E**) Ratios of collagen 1/collagen 3. * *p* < 0.05, ** *p* < 0.01, *** *p* < 0.001. Data indicate mean ± SD (*n* = 4 per group).

**Figure 4 ijms-26-02286-f004:**
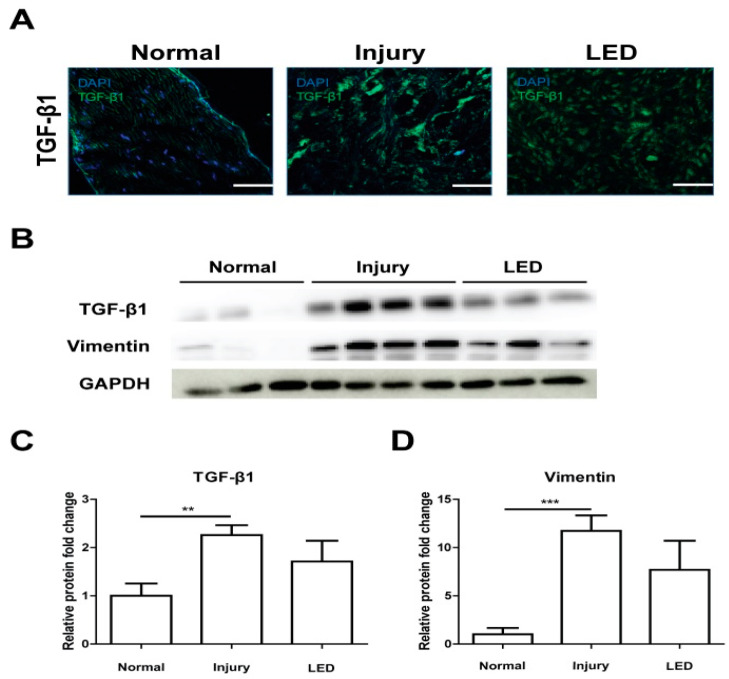
Effects of LED irradiation on myofibroblast formation of injured Achilles tendon. (**A**) Immunostaining of TGF-β1 in Achilles tendon tissue at 2 weeks post-surgery. Nuclei were stained with DAPI. Scale bar = 50 μm. (**B**) Representative image of Western blot analysis of transforming growth factor beta-1 (TGF-β1) and vimentin. (**C**,**D**) Quantification of TGF-β1 and vimentin protein levels. The relative protein levels of TGF-β1 (**C**) and vimentin (**D**) vs. GAPDH were measured. ** *p* < 0.01, *** *p* < 0.005. Data indicate mean ± SD (*n* = 4 per group).

**Figure 5 ijms-26-02286-f005:**
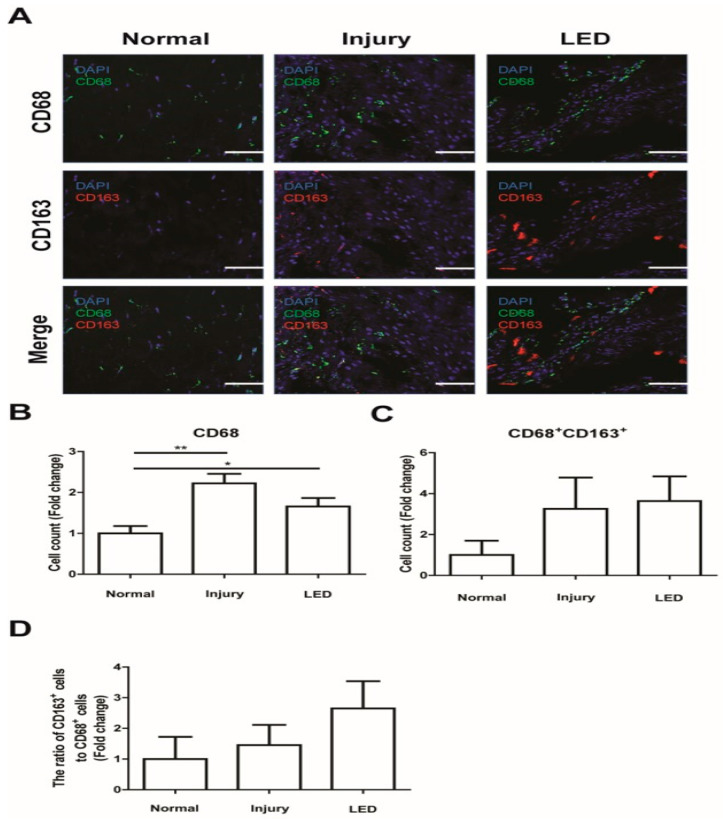
Effects of LED irradiation on macrophage activation in injured Achilles tendon. (**A**) Immunostaining for CD68 and CD163 expression in Achilles tendon tissue at 2 weeks post-surgery. Nuclei were stained with DAPI. Scale bar = 50 μm. (**B**) Quantification of CD68^+^ macrophages in the Achilles tendon tissues. (**C**) Quantification of CD68^+^CD163^+^ M2 macrophages in the Achilles tendon tissues. (**D**) Comparison of the ratio of M2 macrophage (CD68^+^CD163^+^)/pan macrophage (CD68^+^). * *p* < 0.05, ** *p* < 0.01. Data indicate mean ± SD. (*n* = 4 per group).

## Data Availability

The datasets used and/or analyzed during the current study are available from the corresponding author upon reasonable request.

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
