# Peer review of "Efficacy of Light-Emitting Diode-Mediated Photobiomodulation in Tendon Healing in a Murine Model"

_ijms, 2025, doi:10.3390/ijms26052286_

Round 1
Reviewer 1 Report
Comments and Suggestions for Authors
A well-written paper and a good study.
In the methods, be a bit more explicit about the treatment of the mice. ie which were surgically altered; were the healthy controls anaesthetised; was a sham LED treatment administered?
A note that the results reference rats rather than mice.
If the healthy controls were not anaesthetised, how might this influence the results?
Author Response
Reviewer 1
A well-written paper and a good study.
In the methods, be a bit more explicit about the treatment of the mice. ie which were surgically altered; were the healthy controls anaesthetised; was a sham LED treatment administered?
If the healthy controls were not anaesthetised, how might this influence the results?
Answer: Thanks for your thoughtful comments. We have clarified that mice in the healthy control group, which did not undergo surgery, were anesthetized under the same conditions as the othet group to ensure that anesthesi did not influence the experimental outcomes. Likewise, during LED irradiation, the non-irradiated groups (the tendon injury group without LED exposure and the healthy control group) were also placed under the same anesthesia protocol. In addition, to provide a clearer description of the animal experiments, we have enhanced the Materials and Methods section in the manuscript (lines 95-97, lines 117-119) by a more detailed explanation of the Achilles tendon surgery and LED irradiation procedures.
A note that the results reference rats rather than mice.
Answer: We sincerely appreciate your meticulous review, which allowed us to identify and rectify this oversight. We have corrected the terminology in the results section (lines 166-167), ensuring that 'mouse/mice' and 'rat/rats' are accurately distinguished. Thank you again for your valuable feedback.
Reviewer 2 Report
Comments and Suggestions for Authors
Regarding the design of this study, I believe that the dosimetry should be better described, including the time of application of LED therapy. Furthermore, I believe that extrapolating treatments performed on rats to humans, who have different tissue sizes and metabolisms, can sometimes generate results that do not correspond to what occurs clinically in LED therapy in humans.
I believe that the described crushing method does not report whether there was a rupture of the tendon with a gap between two parts of the tendon. I believe that this detail would have to be better described so that we know whether a new union occurs between the sectioned parts or just a recovery of crushed and unruptured ligament tissue.
Author Response
Reviewer 2
Regarding the design of this study, I believe that the dosimetry should be better described, including the time of application of LED therapy.
Answer: Thank you for your valuable comments. Regarding your concern about dosimetry, we would like to clarify that the LED therapy was applied using two wavelengths, 630 nm and 880 nm, simultaneously. The LED irradiation was conducted for 20 minutes at 24-hour intervals over a period of two weeks. The total delivery energy density was 60 J/cm2, as the 630 nm and 880 nm wavelengths were applied simultaneously at 12 J/cm2 and 48 J/cm2, respectively. We have ensured that this information is clearly described in the manuscript (lines 106-116).
Furthermore, I believe that extrapolating treatments performed on rats to humans, who have different tissue sizes and metabolisms, can sometimes generate results that do not correspond to what occurs clinically in LED therapy in humans.
Answer: Thanks for your valuable comment. This study is focused on observation of the effects of LED therapy on the actual recovery process. While we acknowledge the differences between animal models and humans, the mouse model serves as a useful toll for assessing the impact of LED therapy on tissue healing. However, additional studies are required to bridge the gap between preclincal findings and clinical outcomes.
I believe that the described crushing method dose not report whether there was a rupture of the tendon with a gap between two parts of the tendon. I believe that this detail would have to be better described so that we know whether a new union occurs between the sectioned parts or just a recovery of crushed and unruptured ligament tissue.
Answer: Thank you for your comment. We acknowlege the importance of clarifying whether the Achilles tendon injury model involves a complete rupture with a gap or merely a crush injury.
In our study, we performed a transection of the Achilles tendon (2 mm diameter) rather than a simple crush injury. The procedure was as follows: After making a skin incision, we completely transectied the midpoint of the tendon using scissors. A suture with a knotted end was then passed through one side of the served tendon and subsequently through the other side to reconnect the two tendon ends. Finally, the skin incision was suctured.
We have followed the surgical methods with slight modifications to suit our experimental conditions, as described by Watanabe G et al. (Int J Mol Sci. 2023; 24(14):11305) and Zhang J et al. (Nano Res. 2024;17:778-787). To clarify our methodology, we have revised the description of the surgical procedure in the manuscript to provide additional details and included a figure illustrating each step as Supplementary Figure 1 (lines 86-95).